# Exploring Circulating Long Non-Coding RNAs in Mild Cognitive Impairment Patients’ Blood

**DOI:** 10.3390/biomedicines11112963

**Published:** 2023-11-02

**Authors:** Bruna De Felice, Cinzia Coppola, Simona Bonavita, Elisabetta Signoriello, Concetta Montanino, Federica Farinella

**Affiliations:** 1Dipartimento di Scienze e Tecnologie Ambientali, Biologiche e Farmaceutiche, Università degli Studi della Campania “Luigi Vanvitelli”, Via G. Vivaldi 42, 81100 Caserta, Italy; concetta.montanino@unicampania.it; 2Department of Advanced Medical and Surgical Sciences, University of Campania “L. Vanvitelli”, 80138 Naples, Italy; cinzia.coppola@unicampania.it (C.C.); simona.bonavita@unicampania.it (S.B.); elisabetta.signoriello@unicampania.it (E.S.); 3Division of Clinical Pathology, Laboratori Vita S.r.l., Via Sabaudia 19, 04100 Latina, Italy; federica.farinella@vitalab.it

**Keywords:** lncRNAs, MCI, SNHG16, H19, NEAT1, Alzheimer’s disease, mild cognitive impairment

## Abstract

Mild cognitive impairment (MCI) is a transitional clinical stage prior to dementia. Patients with amnestic MCI have a high risk of progression toward Alzheimer’s disease. Both amnestic mild cognitive impairment and sporadic Alzheimer’s disease are multifactorial disorders consequential from a multifaceted cross-talk among molecular and biological processes. Non-coding RNAs play an important role in the regulation of gene expression, mainly long non-coding RNAs (lncRNAs), that regulate other RNA transcripts through binding microRNAs. Cross-talk between RNAs, including coding RNAs and non-coding RNAs, produces a significant regulatory network all through the transcriptome. The relationship of genes and non-coding RNAs could improve the knowledge of the genetic factors contributing to the predisposition and pathophysiology of MCI. The objective of this study was to identify the expression patterns and relevant lncRNA-associated miRNA regulatory axes in the blood of MCI patients, which includes lncRNA-*SNHG16*, lncRNA-*H19*, and lncRNA-*NEAT1*. Microarray investigations have demonstrated modifications in the expression of long non-coding RNAs (lncRNA) in the blood of patients with MCI compared with control samples. This is the first study to explore lncRNA profiles in mild cognitive impairment blood. Our study proposes RNAs targets involved in molecular pathways connected to the pathogenesis of MCI.

## 1. Introduction

Mild cognitive impairment (MCI) is a state considered a “transitional zone” between normal aging and dementia [1]. Morphologically, MCI shows some changes in brain structure stackable to Alzheimer’s disease such as beta-amyloid plaques, tangles of tau proteins, and microscopic clumps of Lewy body proteins, determining alterations such as the decreased size of the hippocampus, increased size of ventricles, and reduced use of glucose in key brain regions resulting in memory impairment. The strongest genetic risk factors for the MCI neurodegenerative cascade are the allelic variant e4 of the human Apolipoprotein E (*APOE*) gene and the systemic inflammatory responses, which are composed of increased levels of soluble tumor necrosis factor receptor 2 (sTNFR2), monocyte chemoattractant protein-1 (MCP-1), and IL-6, and decreased levels of IL-8 [2]. No test currently represents the gold standard confirming that someone has a mild cognitive impairment (MCI). MCI diagnosis is based on mental status tests (Short Test of Mental Status, Montreal Cognitive Assessment (MoCA), or the Mini-Mental State Examination (MMSE)), neurological exams, and lab tests (vitamin B-12, thyroid hormone, and brain imaging), but no none of them identify the MCI state uniquely [3]. Therefore, in recent years, research has increasingly focused on the involvement of circulating non-coding RNAs (ncRNAs) in the development and/or progression of neurodegenerative diseases. Actually, we know that ncRNAs as long non-coding RNA (lncRNA), micro RNAs (miRNAs), small nucleolar RNAs (snoRNAs), and circular RNAs (circRNAs) are key regulators in transcription and post-transcriptional modifications, cell metabolism, proliferation, and apoptosis [4]. Several studies reported that miRNA deregulation contributes to neurodegeneration, as mir-567 [5] or the triad *miR-181a-5p*, *miR-146a-5p*, and *miR-148a-3p* [6] impair neural plasticity and cognitive function in MCI patients advancing to AD. Again, some miRNA families as miR-132 and/or miR-134 were shown to be deregulated in MCI patients, specifically miR-132 for its involvement in tau metabolism through *ITPKB* gene (Inositol-Trisphosphate 3-Kinase B) targeting [7]. LncRNAs, which range in length from 200 nt to over 100 kb, are transcripts that resemble mRNAs in terms of biogenesis, but they differ from mRNAs in that they do not have an open reading frame (ORF) of significant length, or a translation capacity [8]. They are located in subcellular compartments, such as the nucleus, cytoplasm, and foci of cells [9]. LncRNAs are known for their involvement in gene expression regulation, acting as miRNA decoys or by trapping mRNAs in nuclear bodies, and in translation interference, by preventing protein phosphorylation or disrupting ribosome recruitment [10]; but the role of lncRNAs in neurodegenerative diseases, particularly in MCI, is poorly known. The lack of knowledge about this contribution prompted us to study the association between circulating lncRNAs and MCI onset through the investigation of the whole transcriptome. Here, for the first time, microarray investigations have demonstrated modifications in the expression of long non-coding RNAs (lncRNA) in the blood of patients with MCI, comprising lncRNA-*SNHG16*, lncRNA-*H19*, and lncRNA-*NEAT1*.

This research offers a starting point to comprehend the roles of such long non-coding RNAs in MCI from the earliest stages of the disease.

## 2. Materials and Methods

### 2.1. Recruitment of Patients

To profile circulating lncRNAs, we drew blood samples from 10 patients (6 females, 4 males), aged 59.07 ± 18.76 years, and 10 healthy volunteers (control group; 5 male, 5 females), aged 62.25 ± 11.26 years old, recruited from the Neurology Unit of Policlinic Federico II, Naples. The participants in the study were subjected to a neuropsychological evaluation, a dosage of folic acid and vitamin B_12_, a serum diagnosis to exclude syphilis, and an assessment of thyroid function. In addition, a PET scan was performed to determine the eventual presence of amyloid accumulation on four key cortical regions: frontal, lateral temporal, posterior cingulate/precuneus, and the parietal lobes. MCI subjects show objective evidence of cognitive impairment evaluated using a Mini-Mental State Examination (MMSE) score, and an absence of difficulties in functional activities of daily life, dementia, or other cerebrovascular pathology-related cognitive decline or metabolic/endocrine disease [11]. The MCI patients’ and healthy controls’ characteristics are reported in Table 1. After obtaining written informed consent from all participants for the use of their fluid specimens, the entire trial was conducted under the guidelines of The Ethics Committee of the University of Study of Campania “Luigi Vanvitelli” which approved the research (Prot. 12478/20), and the samples were handled according to the guidelines of the Helsinki Declaration.

### 2.2. Blood Collection

For lncRNA quantification, we collected a venous blood sample (4 mL) from each participant, and we isolated PBMCs (peripheral blood mononuclear cells) using an EDTA-K2 anticoagulant tube (VACUETTE^®^), according to the manufacturer’s instructions.

### 2.3. RNA Extraction

Total RNAs were extracted from entire blood samples using Trizol Reagent (Invitrogen, Carlsbad, 5781 Van Allen Way, United States #15596-026) plus an optional DNase digestion step in agreement with the manufacturer’s protocol, and stored at −80 °C before use. The RNA quality and quantity were analyzed using the Nanodrop 2000 (Thermo Fisher Scientific, Waltham, MA USA, 02451), and then total RNA (1 μg) was reverse-transcribed using RT2 First Strand Kit (Qiagen, Tegelen, Hulsterweg 82, The Netherlands).

### 2.4. lncRNA Microarray Profiling

We executed microarray profiling using Human RT^2^ lncRNA PCR Assays (Qiagen). RT^2^ SYBR Green FAST Master mix for qPCR (Qiagen) was used and followed by hybridization to the chip. We then scanned the microarrays using an Agilent microarray scanner directed by GenePix Pro 6.0 software (Axon). The Agilent Feature Extraction 11.5 software elaborated the TIFF images for grid alignment and expression data analysis. The expression data were normalized using quartile normalization and the (RMA) algorithm that was included in the Agilent Feature Extraction 11.5 software. We identified differentially expressed lncRNAs through fold change filtering. The microarray data with threshold values of −2 < fold change < 2 under FDR protection (*p* < 0.05) were selected.

### 2.5. Validation of lncRNA Gene Expression in Blood by Real-Time Quantitative PCR (RT-qPCR)

We performed a RT-PCR quantification of lncRNA expression in duplicate for each sample using lncRNA assays (Applied Biosystems Inc., Foster City, CA, USA), according to the manufacturer’s protocol, and then we evaluated the specificity of the PCR product through the dissociation curve. The ABI Prism 7500 sequence detection system (Applied Biosystems, Foster City, CA, USA) was used to perform RT-PCR. The reactions were performed in a mixture (20 μL) containing 5μL cDNA template, 10 μL 2 × SYBR-Green PCR Mix (Qiagen), and 0.5 μL each of sense and antisense primers. Actin β was used as the internal control. For quantitative results, the expression of each lncRNA was represented as a fold change using the 2^−ΔΔCt^ method, and then analyzed for statistical significance.

### 2.6. Statistical and Bioinformatics Analyses

A two-tailed unpaired *t*-test was performed on normalized delta CT values from both MCI and negative controls, to obtain the fold change of the mean gene expression values between the two groups, and their relative *p*-value. The MCI sample “NMC12” was excluded from the statistical analyses as it represented an outlier, having a Z-score higher in absolute value than three deviation standards. Python v. 3.7.0 was used for the statistical studies, together with its modules numpy [12], SciPy [13], and Pandas [14]. The Matplotlib [15] and Seaborn [16] libraries were used to visualize data and create graphs.

We searched for possible interactors using the web tool RNAInter v4.0 [17,18]. For each lncRNA, the query was set to search for human RNA-associated interactions collected only from experimental literature using strong detection methods. The resulting interaction databases were joined into a single database and processed using Python v. 3.7.0 and its libraries Pandas [14] and Glob [19].

A pathway analysis was performed on the resulting list of lncRNAs and their interactors, using the g:Profiler [20] web-based toolset. The sources selected were for Gene Ontology: GO molecular function and GO biological process; for biological pathways: KEGG, Reactome, and WikiPathways; for regulatory motifs in DNA: miRTarBase. The significance threshold was set to 0.0001, and the maximum size of the set was 800. Network visualization was performed using Cytoscape [21] v.3.10.0. The interaction dataset was imported selecting only the stronger interactions starting from a score of 0.3, adding also a column bearing the expression values for the lncRNAs. G:Profiler results were imported using the add-on Enrichment Map [22] v.3.3.6. To obtain a less dense and more readable network, the FDR Q value cutoff was set to 0.00001, and the edges and connectivity were set to “sparser”. Clusters belonging to common pathways were manually grouped, circled, and labeled. Important nodes and hubs were selected using the add-on cytoHubba, ranked according to their Maximal Clique Centrality score [23].

## 3. Results

We first performed a two-tailed unpaired t-test on the differentially expressed lnc-RNAs, to assess their fold change and corresponding *p*-value. The results, visible in Figure 1, show that the majority of the differentially expressed lncRNAs are significantly overexpressed in MCI patients with respect to controls, except for lncRNA-*NEAT1*, which is significantly under-expressed. By contrast, lncRNAs *MALAT1*, and XIST fold change in expression appear to be non-significant. The top ten lncRNAs with the highest fold change are *DIO3OS*, *LINC00853*, *BOK-AS1*, *CCAT2*, *BCYRN1*, *WT1-AS*, *FENDRR*, *NRON*, and *PANDAR*.

A list of interactors for each lncRNA was downloaded from the RNAInter interactome database with the purpose of building an interaction network. The interactions were filtered to include only those with strong experimental evidence. Interactors include micro-RNAs (miRNAs), lnc-RNAs, mRNAs, transcription factors (TF), proteins, DNA loci, and compounds. The resulting network is displayed in Figure 2a. The nodes representing the lncRNAs which are the object of our study are colored according to their fold change, while the interactors are in grey. Among the various interactions present in the network, it is worth highlighting the indirect interaction that may occur between the third highest expressed lncRNA, *BOK-AS1*, and the only downregulated lncRNA, *NEAT1*, via their common interactors, the RNA-binding proteins *NONO* and *SPFQ* (visible in Figure 2b).

Interestingly, the overexpression of *NEAT1* was linked to Alzheimer’s Disease (AD) and memory impairment in many studies [24,25,26], while its knockdown in mice leads to decreased memory deficit and increased dendritic spine density [27], and it has been suggested as plasma biomarker for AD progression [28].

Subsequently, we proceeded to look for lncRNAs that could act as hubs of connectivity among the network, so we used cytoHubba to obtain a list of the top 10 lncRNAs ranked for their maximal clique centrality (MCC) score (Figure 3, Table 2). The list includes, from the highest to the lowest score, *SNHG16*, *MALAT1*, *H19*, *HOTAIR*, *MEG3*, *NEAT1*, *TUG1*, *XIST*, *GAS5*, and *UCA1*. It is worth noting that the most researched and well-characterized lncRNAs will tend to have a high number of known interactors, and on the contrary, less researched lncRNAs will show a lower number of interactions.

The next step was to use the expanded list of lncRNAs and their interactors, to perform a pathway analysis using G:Profiler. According to the results, shown in Figure 4, we obtained several networks; single nodes and networks belonging to related pathways were manually grouped and labelled to enhance comprehensibility. The most populated group includes nodes characterized predominantly by microRNAs. Encapsulated within these nodes are lncRNAs and their interactors, which may be regulated or interact with these miRNAs. A second prominent cluster represents signaling pathways involved in growth factor response; this cluster is linked, via a single edge, to a smaller cluster related to cell cycle progression and G1-to-S phase transition. This same cluster is also linked via seven edges to another sizable network, which includes pathways involved in tissue morphogenesis, angiogenesis, and cellular migration and/or adhesion. Other significant networks include pathways involved in the regulation of cell death, stress and damage response, positive regulation of kinase activity, regulation of transcription, organelle organization, hormone response, and regulation of protein metabolism. At last, there are two independent nodes representing interleukin 4 and 13 signaling, and MECP2-associated pathways.

## 4. Discussion

Mild cognitive impairment (MCI) is regarded as a prodrome of dementia; in fact, around half of the patients diagnosed with MCI will develop dementia within 3 years. There are three kinds of MCI: non-amnestic, amnestic, and multi-domain impairment; not all of them develop into Alzheimer’s disease (AD). The prevalence of MCI in the community is difficult to determine, but different studies reported a percentage of 3–25%, dependent on mean age and other risk factors (smoking, diabetes, metabolic syndrome, etc.) [29]. MCI diagnosis is based on the same techniques used for AD. After an initial Mini-Mental Status Examination (MMSE), patients are investigated through neuroimaging techniques and amyloid beta (Aβ_42_), total tau, and phosphorylated tau detection in the cerebrospinal fluid (CSF). Moreover, the epsilon 4 allele of the apolipoprotein gene (APOE-e4) confers a risk of MCI [30]. MCI can be considered as a “window” in which intervention and delaying the onset of dementia may be possible, and physicians are always searching for a select few biomarkers that would serve as the gold standard for early MCI diagnosis. For this study, we enrolled a group of 10 MCI patients whose diagnosis was reached according to the IWG-2 criteria [5].

LncRNAs are ncRNAs (~200 nt) that are not capable of coding for proteins. They have a 5′ cap structure, multiple exons, 3′ polyadenylated tails, and are spliced in a way similar to mRNAs. Not only DNA methylation and histone modification, but also a combination of ncRNAs and transcription factors are involved in genome epigenetic modifications. Several studies on the role of lncRNAs suggest that their dysregulation could trigger neuronal death via still unexplored RNA-based regulatory mechanisms [31].

Herein, we observed lncRNAs in a small, well-characterized population of MCI patients. Little is known about lncRNAs in neurodegeneration, especially in the MCI state. However, lncRNAs in the MCI state can be significant for their regulatory role in biological processes and potential role as an early biomarker for diagnosis.

In this study, we have found a group of significantly deregulated lncRNAs in MCI patients compared to healthy controls. We then expanded this list with the experimentally known interactors for each lncRNA, and generated an interactome network and a pathway network, in order to have a broad overview of the processes that may be involved in the pathogenesis of MCI, and possibly, in its progression to Alzheimer’s Disease.

Given the complexity of the results, we choose to focus our observations on three promising lncRNAs: *SNHG16*, *H19*, and *NEAT1*.

Among our list of deregulated lncRNAs, *SNHG16* has the highest MCC score, given the interactions it exerts with a plethora of miRNAs, suggesting that it may play a crucial role in coordinating a particular biological process or function, within a larger network of genes involved in MCI. Distinctly, lncRNA *SNHG16*, observed to be upregulated in MCI, showcased a neuroprotective role in both rat cells treated with dexmedetomidine, and hESC-derived neurons treated with ketamine. Its involvement in the *mir-10b-5p*/BDNF and NeuroD1 pathways, respectively, raises pivotal questions about its broad-spectrum functionality in neuroprotection [32,33]. Furthermore, the protective attributes of *SNHG16*, which intervenes against oxygen glucose deprivation/reoxygenation-induced apoptosis in hBMECs, illustrate the intricate and multifaceted roles this lncRNA plays in cellular responses to stress and damage [34]. Due to its significance, it merits further investigation to better understand its role in the development and progression of MCI.

If we set aside *MALAT1*, which as our data indicate, has expression levels not significantly different from those of the control group, *H19* is the next lncRNA with the highest MCC score. H19 stands out as one of the earliest imprinted lncRNAs identified, with a pivotal role in both physiological and pathological contexts. For instance, it has been observed to be involved in neuroinflammation upon ischemic stroke (IS), where *H19* reportedly competes with *miR-138-5p* to stimulate the NF-κB pathway, thus augmenting the release of pro-inflammatory cytokines [35]. The observed shift from the pro-inflammatory M1 phenotype to the anti-inflammatory M2 phenotype upon *H19* knockdown further underlines its importance in inflammatory modulation [36,37]. Although the precise role of *H19* in MCI has yet to be assessed and its role in Alzheimer’s Disease remains elusive, it was observed that in Aβ25-35-induced PC12 cells, lncRNA *H19* negatively regulates the expression of the pro inflammatory cytokine HMGB1 by targeting miRNA *miR-129* and acting as a sponge, consequently promoting inflammation, while its knockdown ameliorates inflammation and neurological function in rat models, making it a promising pharmacological target [38].

Finally, our results show *NEAT1* to be significantly less expressed in MCI patients than in controls. As mentioned before, the identified potential interaction between lncRNAs *BOK-AS1* and *NEAT1* underscores the complex regulatory network of *NEAT1*’s associations with Alzheimer’s Disease, both in terms of memory impairment and its knockdown benefits, further consolidating its significance in neurological contexts [24,25,26,27,28].

In the G:Profiler pathway analysis, a prominent cluster was identified, characterized predominantly by miRNAs. The prominence of this miRNA-centric cluster may indicate a potential significance of miRNA-mediated regulation in the context of MCI. In fact, each miRNA node within the cluster can potentially regulate multiple lncRNAs and lncRNA interactors, implying a complex regulatory network that warrants further exploration. For example, while *miR-34a-5p* has been shown to be upregulated in AD patients’ brains [39], it is also known to target the 3′UTR region of tau, reducing its expression [40]. In contrast to its upregulation in AD, *miR-34a-5p* is downregulated in mouse brains following traumatic brain injury (TBI) [41]. This downregulation could lead to an increased expression of tau, among other effects, potentially exacerbating the negative outcomes associated with TBI. The contrasting expression profiles of *miR-34a-5p* in AD and TBI highlight the intricate and multifaceted roles miRNAs play in neurodegenerative and neurological disorders.

The second most prominent group was composed of various signaling pathways related to the regulation of cell growth and cell cycle progression. Among the pathways of this group, AKT, PTEN, and ROCK2 were recurrently present. The PI3K/AKT pathway is known to be pivotal for brain development, inhibiting autophagy, enhancing neurite extension, and regulating neuronal synaptic plasticity [42]. PTEN, a phosphatase, negatively regulates this pathway by converting PI(3,4,5)P3 to PI(4,5)P2 [43]. ROCK2 is found in the brain, and its expression can increase along with age and due to neurodegenerative disorders [44]. This protein phosphorylates PTEN, thus inhibiting the PI3K/AKT pathway and promoting apoptosis [45]. It is worth noting that *NEAT1* is present in four nodes among this group, as shown in more detail in Table 3. It has been observed in a rheumatoid arthritis study that *NEAT1* is able to activate the ROCK2 and WNT pathway by inhibiting *miR-144-3p* [46]. If such interaction also occurred in neurodegeneration, this would possibility implicate a suppression of the PI3K/AKT signaling pathway, that in turn would lead to an increase of neuronal apoptosis, a reduction of neurite growth and synaptic plasticity, and a suppression of pro-survival pathways. Yet, as stated above, *NEAT1* was significantly less expressed in MCI patients compared to controls. Considering our results, we might speculate that the downregulation of *NEAT1* in MCI might represent an early protective or compensatory response, aimed to counteract or to slow the progression of neurodegenerative changes. However, more research will be required to validate this theory.

Another relevant group refers to cellular responses to stress and damage response, in particular to hypoxia. In MCI patients, alterations in cerebral blood flow have been observed which are correlated with a high level of amyloid-β deposition, possibly causing local hypoxia [47]. Furthermore, genes responsive to low oxygen levels seem to be activated in MCI before the development of AD. Similarly, mouse brains exhibited early reactions to mild oxygen shortage, reflecting changes seen in MCI. Extended low-oxygen reactions in the brain were linked with the activation of the PI3K/AKT/mTOR pathways in specific brain vessels [48]. Overall, vascular changes seem to play a critical role in dementia’s early stages, acting as a response to reduced oxygen levels even before AD onset.

Lastly, it is worth drawing attention to the cluster referred to as “adhesion and development”; this group includes adhesion molecules such as ICAM2 (intracellular adhesion molecule 2) and CDH5 (cadherin 5) that are part of the brain endothelial cells (BECs), which form the primary layer of the blood–brain barrier. In the Atherosclerosis Risk in Communities study, these proteins correlated with dementia risk over 25 years, suggesting that adhesion molecules may be valuable in predicting neurovascular outcomes and assessing blood–brain barrier integrity [49].

Altogether, pathway analysis results indicate that the dysregulated lncRNAs, and their associated interactors, could play pivotal roles in the multifaceted pathogenesis of MCI. Through their interactions, they can potentially influence a range of crucial biological processes, ranging from signaling pathways and stress responses to hypoxia, inflammation, and cellular adhesion, as well as the regulation of transcription. Recognizing these complex connections may help in deepening our understanding of MCI, and possibly pave the way for potential therapeutic targets and intervention points to combat its progression.

## 5. Conclusions

To summarize, we found lnc-RNAs that are differentially expressed in MCI patients compared to controls, and built a network including their experimentally validated interactors. Among these, *SNHG16*, *H19*, and *NEAT1* were part of the lncRNAs forming the highest number of connections. Moreover, their intricate roles and interactions in various neurological conditions suggest that their implication in MCI may be plausible. Finally, we provided a comprehensive overview of potentially significant pathways involved in MCI, offering new starting points for future investigations in the field.

As our understanding of these lncRNAs continues to evolve, it is evident that they could present promising avenues for MCI therapeutic interventions and diagnostic strategies. However, future studies elucidating their mechanistic pathways are imperative for translating these findings into clinical applications, and a prospective study is needed to assess an eventual role of these lncRNAs in the MCI-to-AD conversion.

## Figures and Tables

**Figure 1 biomedicines-11-02963-f001:**
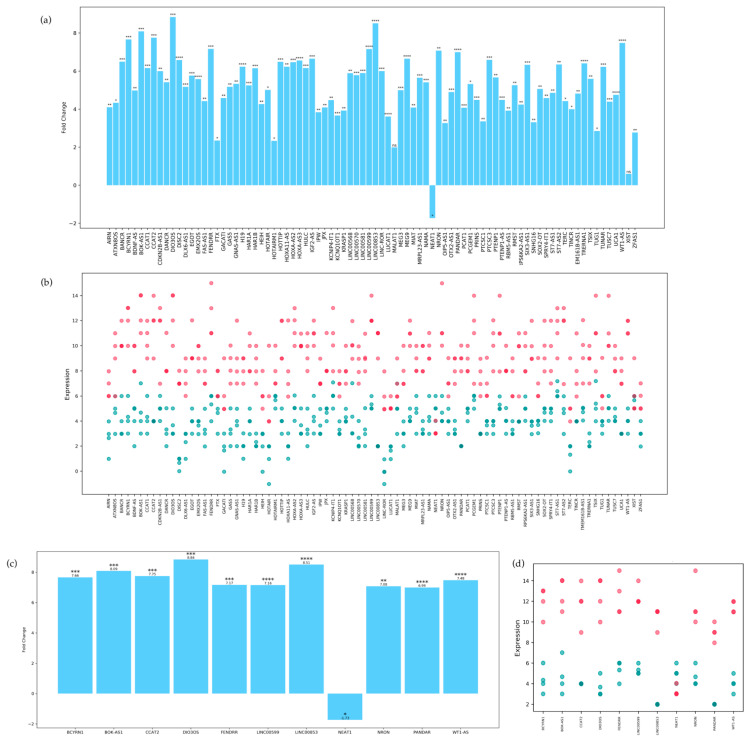
(**a**) Column histogram showing the 84 lncRNAs on the x axis and their respective fold change on the y axis. Significance is attributed in the following way: ns for *p*-values above 0.05; * for *p*-values between 0.05 and 0.01; ** for *p*-values between 0.01 and 0.001; *** for *p*-values between 0.001 and 0.0001; and **** for *p*-values smaller than 0.0001. (**b**) Scatterplot representing each lncRNA on the x axis and their respective expression on the y axis. Each dot represents the expression in a single sample. MCI patients are colored magenta while control patients are cyan. A deeper color represents two or more patients having a similar expression level. (**c**) Column histogram showing only the top 10 lncRNAs with the highest fold change, plus *NEAT1*. (**d**) Scatterplot showing the expression distribution for every sample of the top 10 lncRNAs with the highest fold change, plus *NEAT1*.

**Figure 2 biomedicines-11-02963-f002:**
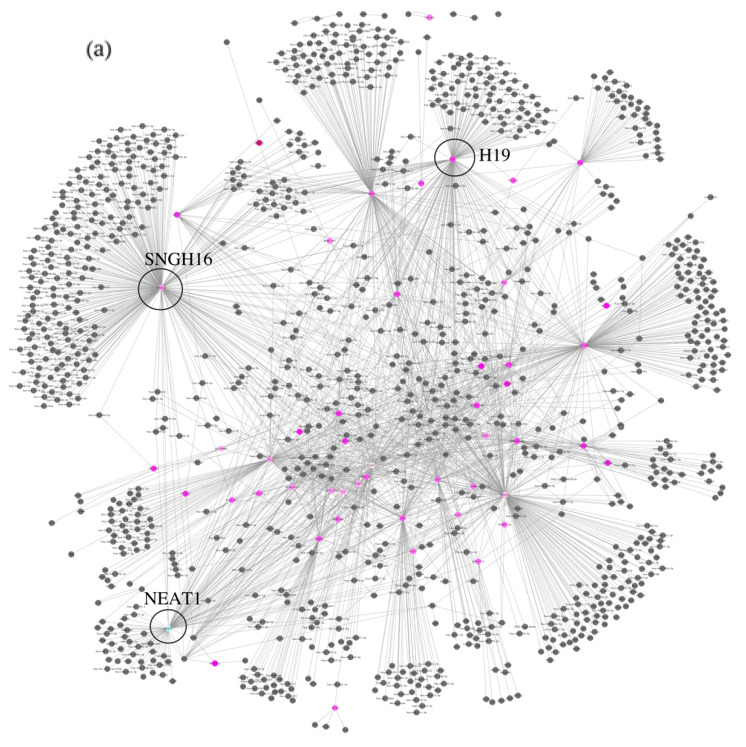
(**a**) Interaction network. The nodes representing the lncRNAs taken into consideration are colored differently based on their expression. Positive fold change is represented by pink color, and the deeper the pink, the higher the fold change. White color corresponds to zero-fold change, while a negative fold change is represented by blue color. Interactors for which the expression is unknown are represented by grey color. (**b**) Overall view of the interaction network and zoomed-in detail of the indirect interaction between *NEAT1* and *BOK-AS1*. The common interactors *NONO* and *SFPQ* are highlighted in yellow.

**Figure 3 biomedicines-11-02963-f003:**
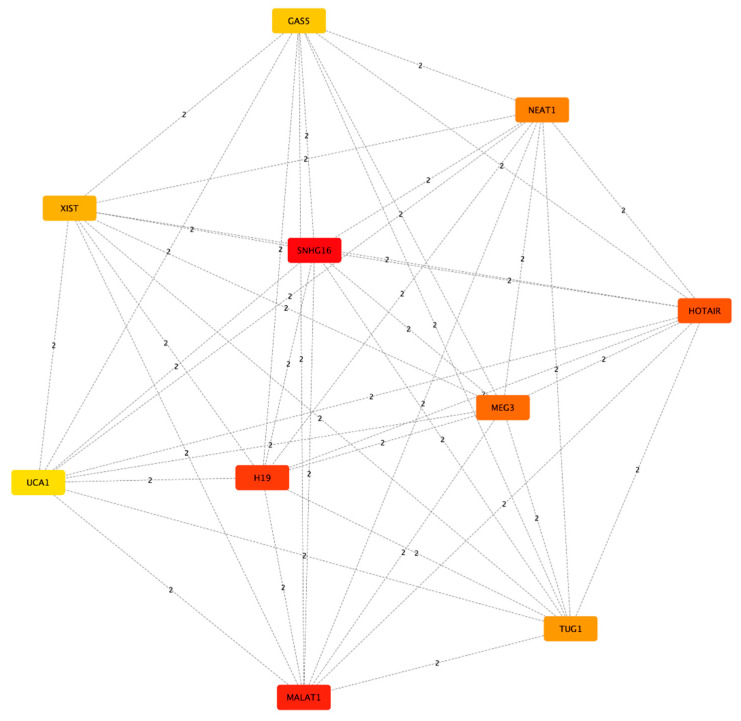
Subnetwork created from the interaction network representing the top 10 lncRNAS having the higher MCC score. Nodes are colored according to the score, with red nodes having the highest score and yellow the lowest score. Numbers within the edges indicate the length of the path between two nodes.

**Figure 4 biomedicines-11-02963-f004:**
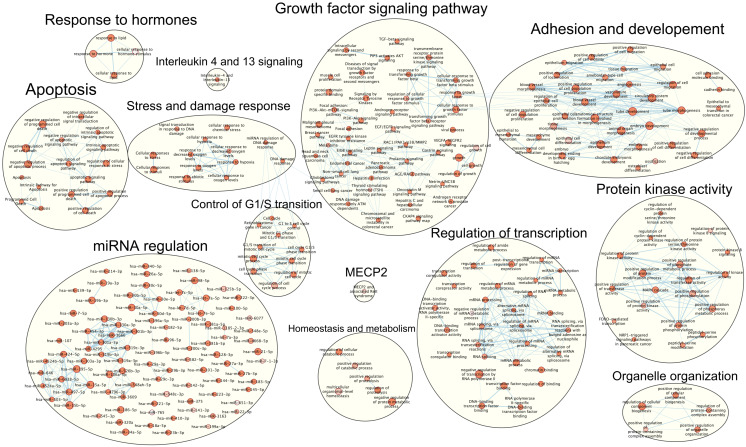
Network resulting from the pathway analysis. Each node’s width varies in direct proportion to the number of its associated lncRNAs or interactors. LncRNA- or interactor-sharing pathways are connected by blue edges, with the thickness of the edges representing the quantity of shared lncRNAs or interactors.

**Table 1 biomedicines-11-02963-t001:** Summary statistics for patients with MCI and control patients.

Clinical Characteristics	Value	MCI Patients	Healthy Controls
Age at our observation (years)	Mean ± STD	59.07 ± 18.76	62.25 ± 11.26
	Range	49–65	52–66
Age at symptom onset (years)	Mean ± STD	55.17 ± 10.35	-
	Range	54–63	-
Sex	Male	4	5
	Female	6	5
Mini-Mental State Examination (MMSE) score	-	22.48 (±2.06)	28.47 (±1.93)
Other cerebrovascular pathology	-	None	None
Metabolic/endocrine disease	-	None	None

**Table 2 biomedicines-11-02963-t002:** Top 10 lncRNAs ranked by maximal clique centrality score.

Rank	Name	Score
1	*SNHG16*	281.0
2	*MALAT1*	202.0
3	*H19*	174.0
4	*HOTAIR*	153.0
5	*MEG3*	135.0
6	*NEAT1*	102.0
7	*TUG1*	95.0
8	*XIST*	84.0
9	*GAS5*	78.0
10	*UCA1*	63.0

**Table 3 biomedicines-11-02963-t003:** List of pathways including lncRNA-*NEAT1*.

Description	Shared Name
post-transcriptional regulation of gene expression	GO:0010608
positive regulation of cell population proliferation	GO:0008284
regulation of epithelial cell proliferation	GO:0050678
epithelial cell proliferation	GO:0050673

## Data Availability

The data presented in this study are available on reasonable request from the corresponding author.

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
