# Peer review of "Exploring Circulating Long Non-Coding RNAs in Mild Cognitive Impairment Patients’ Blood"

_biomedicines, 2023, doi:10.3390/biomedicines11112963_

Round 1
Reviewer 1 Report
Comments and Suggestions for Authors
Authors presented the results of changes in circulating long non-coding RNAs in blood samples from MCI in comparison to normal.
It would be good to provide the few blood biomarkers to support the validity of the MCI and normal samples, even though the results are focusing on circulating long non-coding RNAs.
In addition, multiple ontology connections with leading genes of circulating long non-coding RNAs were presented. But there are too many and heterogeneous, can authors try to organize and funnel them to the few pathways, instead mentioning SNHG16, H19, and NEAT1 as 3 promising genes? Here, blood biomarkers could help in funneling the pathogenicity.
Can authors segregate the converters and non-converters from MCI to AD using the results? If not, authors need to mention the limitations of the study in the discussion section.
Comments on the Quality of English LanguageThere are TOO MANY run-on sentences, which need to be revised.
Author Response
Comments from Reviewer 1
Comment 1.1: Authors presented the results of changes in circulating long non-coding RNAs in blood samples from MCI in comparison to normal.
Answer 1.1: Thank you for summarizing the focus of our study. We appreciate your time in reviewing our manuscript and for the insightful comments provided, which will be addressed in detail below.
Comment 1.2: It would be good to provide the few blood biomarkers to support the validity of the MCI and normal samples, even though the results are focusing on circulating long non-coding RNAs.
Answer 1.2: We acknowledge the importance of presenting an extensive assessment to support the validity of our MCI and normal samples. Indeed, in addition to the extensive neuropsychological evaluation, we performed a study of thyroid function, serum diagnosis for syphilis, the dosage of vitamin B12 and folic acid, and a PET scan to visualize an eventual accumulation of amyloid proteins in the brain. To address your observation and the omission in our initial submission, we will expand our "Material and Methods" section to include more comprehensive informations about the specific blood biomarkers analyzed, and the imaging techniques employed. (Chapter 2, Materials and Methods, lines 77-81)
Comment 1.3: In addition, multiple ontology connections with leading genes of circulating long non-coding RNAs were presented. But there are too many and heterogeneous, can authors try to organize and funnel them to the few pathways, instead mentioning SNHG16, H19, and NEAT1 as 3 promising genes? Here, blood biomarkers could help in funneling the pathogenicity.
Answer 1.3: Following Reviewer’s suggestions, we:
- reorganized the heterogeneous data from the pathway analysis figure to present clusters that belong to a few significant common pathways. These pathways have been annotated with distinct, easily discernible labels. (Chapter 3, Results, lines 207-219, and Figure 4, line 222)
- we expanded the discussion section to provide a more detailed examination of select pathways. (Chapter 4, Discussion, Figure 4, lines 294 - 352)
Yet we maintain our focus on SNHG16, H19, and NEAT1 due to their relevance as potential biomarkers in our study. We would like to thank Reviewer 1 as the suggested changes offer a clearer, more balanced presentation of our findings.
Comment 1.4: Can authors segregate the converters and non-converters from MCI to AD using the results? If not, authors need to mention the limitations of the study in the discussion section.
Answer 1.4: Thank you for pointing this out. At this stage of the study, we are unable to differentiate the potential converters from MCI to AD. We concur with the need to address this limitation in the discussion, so we added a statement in chapter 4 (Discussion, lines 363-365).
Comment 1.5: There are TOO MANY run-on sentences, which need to be revised.
Answer 1.5: We agree with your observation regarding the run-on sentences in the manuscript. We have thoroughly reviewed the document and revised the problematic sentences for better clarity and coherence. We believe these revisions will enhance the readability of the paper. (Numerous corrections have been made throughout the manuscript and, for clarity, all changes are highlighted in yellow)
Comments from Reviewer 1
Comment 1.1: Authors presented the results of changes in circulating long non-coding RNAs in blood samples from MCI in comparison to normal.
Answer 1.1: Thank you for summarizing the focus of our study. We appreciate your time in reviewing our manuscript and for the insightful comments provided, which will be addressed in detail below.
Comment 1.2: It would be good to provide the few blood biomarkers to support the validity of the MCI and normal samples, even though the results are focusing on circulating long non-coding RNAs.
Answer 1.2: We acknowledge the importance of presenting an extensive assessment to support the validity of our MCI and normal samples. Indeed, in addition to the extensive neuropsychological evaluation, we performed a study of thyroid function, serum diagnosis for syphilis, the dosage of vitamin B12 and folic acid, and a PET scan to visualize an eventual accumulation of amyloid proteins in the brain. To address your observation and the omission in our initial submission, we will expand our "Material and Methods" section to include more comprehensive informations about the specific blood biomarkers analyzed, and the imaging techniques employed. (Chapter 2, Materials and Methods, lines 77-81)
Comment 1.3: In addition, multiple ontology connections with leading genes of circulating long non-coding RNAs were presented. But there are too many and heterogeneous, can authors try to organize and funnel them to the few pathways, instead mentioning SNHG16, H19, and NEAT1 as 3 promising genes? Here, blood biomarkers could help in funneling the pathogenicity.
Answer 1.3: Following Reviewer’s suggestions, we:
- reorganized the heterogeneous data from the pathway analysis figure to present clusters that belong to a few significant common pathways. These pathways have been annotated with distinct, easily discernible labels. (Chapter 3, Results, lines 207-219, and Figure 4, line 222)
- we expanded the discussion section to provide a more detailed examination of select pathways. (Chapter 4, Discussion, Figure 4, lines 294 - 352)
Yet we maintain our focus on SNHG16, H19, and NEAT1 due to their relevance as potential biomarkers in our study. We would like to thank Reviewer 1 as the suggested changes offer a clearer, more balanced presentation of our findings.
Comment 1.4: Can authors segregate the converters and non-converters from MCI to AD using the results? If not, authors need to mention the limitations of the study in the discussion section.
Answer 1.4: Thank you for pointing this out. At this stage of the study, we are unable to differentiate the potential converters from MCI to AD. We concur with the need to address this limitation in the discussion, so we added a statement in chapter 4 (Discussion, lines 363-365).
Comment 1.5: There are TOO MANY run-on sentences, which need to be revised.
Answer 1.5: We agree with your observation regarding the run-on sentences in the manuscript. We have thoroughly reviewed the document and revised the problematic sentences for better clarity and coherence. We believe these revisions will enhance the readability of the paper. (Numerous corrections have been made throughout the manuscript and, for clarity, all changes are highlighted in yellow)

Reviewer 2 Report
Comments and Suggestions for Authors
Bruna De Felice etc. aimed to uncover expression patterns and associations between specific long non-coding RNAs (lncRNAs) and microRNAs (miRNAs) in the blood of Mild Cognitive Impairment (MCI) patients, focusing on lncRNAs SNHG16, H19, and NEAT1. Through microarray analysis, the study revealed altered lncRNA expression in MCI patients compared to healthy individuals, shedding light on previously unexplored lncRNA profiles in MCI blood samples. The research identified several lncRNAs as potential regulators of molecular pathways related to MCI's development and progression to Alzheimer's Disease.
Minor comments:
1. In Table 1, the range of ‘Age at symptoms onset (years)’ seems not correct. It is the same as the above row ‘55.17 ±10.35’. Also suggest renaming the Value of ‘Mean’, as it should be ‘Mean ± STD’ or ‘‘Mean ± SE’.
2. In Figure 1, panels (d) and (e), the order of lncRNAs on the x-axis is inconsistent. NEAT1 is the last one in (d), but not the last one in (e). I suggest keeping the same order.
3. In Figure 2, suggest labeling the name of some important lncRNA in the figure. I can’t tell the results described in the paper from this figure.
“Among the 166 various interactions present in the network, it is worth highlighting the indirect interac-167 tion that may occur between the third highest expressed lncRNA BOK-AS1 and the only 168downregulated lncRNA NEAT1, via their common interactors, the RNA-binding proteins 169 NONO and SPFQ.”
4. The resolution of Figure 4 is too low, need to make it clear.
Author Response
Comments from Reviewer 2
Comment 2.1: Bruna De Felice etc. aimed to uncover expression patterns and associations between specific long non-coding RNAs (lncRNAs) and microRNAs (miRNAs) in the blood of Mild Cognitive Impairment (MCI) patients, focusing on lncRNAs SNHG16, H19, and NEAT1. Through microarray analysis, the study revealed altered lncRNA expression in MCI patients compared to healthy individuals, shedding light on previously unexplored lncRNA profiles in MCI blood samples. The research identified several lncRNAs as potential regulators of molecular pathways related to MCI's development and progression to Alzheimer's Disease.
Answer 2.1: We would like to thank Reviewer 2 for the time and effort dedicated to reviewing our manuscript and for the insightful observations provided, which are addressed below in detail.
Comment 2.2: Minor comments:
- In Table 1, the range of ‘Age at symptoms onset (years)’ seems not correct. It is the same as the above row ‘55.17 ±10.35’. Also suggest renaming the Value of ‘Mean’, as it should be ‘Mean ± STD’ or ‘‘Mean ± SE’.
Answer 2.2: We apologize for this oversight in our manuscript. Upon re-examination, we realized that there was an error in entering the data into Table 1. We appreciate your understanding and have taken steps to rectify this mistake in the revised version of the table. Moreover, we renamed the Mean as “Mean ± STD” according to your suggestion and removed an empty line at the end of the table. (Chapter 2, Materials and Methods, Table 1, line 91)
Comment 2.3: 2) In Figure 1, panels (d) and (e), the order of lncRNAs on the x-axis is inconsistent. NEAT1 is the last one in (d), but not the last one in (e). I suggest keeping the same order.
Answer 2.3: We agree with the inconsistency in the order of lncRNAs on the x-axis. We would like to kindly note that Figure 1 has panels from (a) to (d), and based on your comment, we assume you are referring to panels (c) and (d). We appreciate your suggestion, and we ordered the lncRNAs in panel (c) in alphabetical order to be consistent with the other panels (Chapter 3, Results, Figure 1, line 157).
Comment 2.4: 3) In Figure 2, suggest labeling the name of some important lncRNA in the figure. I can’t tell the results described in the paper from this figure.
“Among the 166 various interactions present in the network, it is worth highlighting the indirect interac-167 tion that may occur between the third highest expressed lncRNA BOK-AS1 and the only 168downregulated lncRNA NEAT1, via their common interactors, the RNA-binding proteins 169 NONO and SPFQ.”
Answer 2.4: We concur with Reviewer 2 that we should provide a more comprehensive visual representation and make the findings discussed in the text easier to correlate with the figures. In response to this suggestion, we have labeled the three most pivotal lncRNAs directly within the figure to ensure clarity and direct correlation with our described results. Furthermore, recognizing the importance of visualizing the described interactions, we have added an additional panel that delineates the indirect circuit of interaction as discussed in the manuscript. (Chapter 3, Results, Figure 2 (a) and (b), lines 178-179)
Comment 2.5: 4) The resolution of Figure 4 is too low, need to make it clear.
Answer 2.5: In response to this observation, we'd like to clarify that the image submitted has a very high resolution (the file is more than 200MB), however, we understand that the complexity of the network with such a significant number of nodes, necessitates zooming in to clearly read the node labels. An increase in label font size, though initially considered, unfortunately leads to an overlap, making the visual representation even more chaotic.
However, we see the point and agree with Reviewer 2 that it is necessary to improve the clarity of the image. To address this, we took a two-pronged approach: firstly, we regrouped clusters belonging to common pathways and annotated them with distinct, readable labels. Secondly, we marginally increased the size of the node labels to enhance their readability. Although zooming may still be required for a crystal-clear view, this revised representation should provide a more comprehensible overview. (Chapter 2, Materials and Methods, lines 145-146; Chapter 3, Results, Figure 4, line 222)
Once again, we would like to thank the Editor and all the Reviewers, your feedback has been instrumental in enhancing the comprehensibility of our work, and we sincerely appreciate it.
We look forward to hearing from you in due time regarding our submission and to respond to any further questions and comments you may have.

Round 2
Reviewer 1 Report
Comments and Suggestions for Authors
Authors revised the manuscript to be accepted.